# Kinetics of Lignin Removal from the Lignocellulosic Matrix after Ozone Transportation

Khurram Shahzad Baig 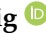

Department of Chemical Engineering, University of Wah, Wah 47040, Pakistan; khurram.shahzad@wecuw.edu.pk

**Abstract:** This study presents a new method to remove lignin from wheat straw (lignocellulosic) using the ozonation technique. Lignocellulosic material is a complex biopolymer composed of cellulose, hemicellulose and lignin. Apart from lignin, which acts as a chemical resistant, lignocellulosic is the main resource of cellulose and hemicellulose sugars. The ozonation reaction takes place in a two-phase solid–gas fluidization stainless steel reactor. The focus of this paper is to investigate the kinetics that govern lignin removal from lignocellulosic material after ozonation treatment. The kinetics of lignin removal did not agree with the experimental data until the suggested model is modified to a pseudo-second-order. The results showed that at a higher ozone supply of 150 mg min$^{-1}$, the surface reaction and intra-particular diffusion were the most significant factors to remove the lignin. Moreover, at a lower ozone supply of 30 mg min$^{-1}$, the intra-particular diffusion was the only contributor towards lignin removal.

**Keywords:** ozonation mechanism; delignification; cellulose to methane; particle diffusion; pseudo-second-order

## 1. Introduction

In the last twenty years, there has been a huge growth in the consumption of liquid fuels rather than fossil fuels. Liquid fuels are generally derived from lignocellulosic biomass wastes that are used to produce energy. They are originally found in the agricultural or animal substances. The sources of biomass are many, including but not limited to, forest or wood residues, wastes from foodstuff processing and livestock farming, or wastes from water treatment plants [1]. The main ingredients of agricultural residues are cellulose, hemicellulose and lignin. The disposition of these components in the structure is that lignin occupies the spaces in the cell wall, which mediates the cellulose and hemicellulose. Explaining the nature for lignocellulosic in terms of chemistry, lignin is covalently bonded to hemicellulose and cross-linked with cellulose. In general, lignin acts as a hindrance to the biofuel (bioethanol) production process, which is produced from cellulose and hemicellulose only, and therefore requires removal by pretreatments.

The literature shows that there are four sets of pretreatments, grouped into physical, chemical, biological, and physiochemical [2]. The major principle of physical pretreatment is to reduce the particle size, to enhance the surface area of a substrate and degrade the polymer chains and crystallinity [3]. This turns out to be more effective and trouble-free for the downstream processes [4] The major drawbacks lie in the amount of investment in mechanical equipment and high-energy consumption [5]. Additionally, chemical pretreatment is a good alternative with respect to the physical pretreatment, which includes treatments with alkali, acids and ionic liquids, or deep eutectic solvents. However, the disadvantage of the chemical pretreatment is the use of expensive, non-environmentally friendly and sometimes highly flammable solvents [6]. Moreover, the biological pretreatment, including cellular and enzymatic, is a promising eco-friendly option with lower energy consumption and requires no inhibitor during the process. On the other hand, ammonia fiber explosion (AFEX), $CO_2$ or steam explosion (SE), liquid hot water (LHW), wet oxidation by hydrogen

peroxide or ozonation are all considered physio-chemical pretreatments, and each of them has its pros and cons. For many years, ozonation pretreatment has been a topic of interest for many researchers [7–12]. This may be attributed to the fact that ozone produces wastes that are less hazardous to the environment compared to conventional pretreatments. Nevertheless, to the best of our knowledge, no one has studied the reaction between ozone and wheat straw.

Based on our earlier findings, it is better to moisten the wheat straw before the start of ozonation because the reaction was very slow with the dry wheat straw, and this may lead to issues related to ozone decomposition. We presumed that the absorbed water on wheat straw causes swelling and improves ozone accessibility to the inner cells. In this paper, we provide a novel technique that is simple and environmentally friendly. The proposed technique is a step-by-step treatment procedure that involves the use of sodium hydroxide (NaOH) solution to remove the natural waxy layer on the wheat straw surface and let the ozone interact with the lignin. Our preliminary work showed if the surface is not cleaned, there will be no reaction of ozone. The main purpose of this technique is to increase the accessibility of ozone for reaction, which leads to an easy lignin removal. It also increases the cellulose exposure to enzymatic hydrolysis and favors benign byproducts.

The study goal is to understand the kinetics governing lignin removal from wheat straw after ozonation pretreatment. Based on a vigorous literature survey, the kinetics of lignin removal using pure oxygen has been widely studied in the paper and pulp industry. In fact, the low contents of lignin, the homogenous distribution of lignin and the gas reactant's easy access to lignin in pulp helped both the first-order kinetic model and the pseudo-first-order kinetic model to describe the kinetics of delignification. Yun reported the removal of lignin from pulp by oxygen and caustic treatments [13]. The delignification reaction rate was a first-order type with residual lignin contents and without hexuronic acid. Roncero et al. investigated the kinetics of ozone bleaching on dried materials using the oven and at room temperature [14]. The ozonation took place in a reactor volume of 0.5 L, perfect mixing was assumed, and the amount of ozone consumed by pulp was tracked every second. In addition, an ozone diffuser was used to provide a consistent and uniform supply of ozone to the fiber suspension. The mass flow rate of ozone into the reactor was 85 mg/min, the flow rate was controlled by a valve and measured by an automatic UV detector. The results showed that the lignin degradation followed first-order kinetics. Other studies have addressed the lignin model to better understand the dynamics of the delignification reaction. For instance, Kishimoto and Sano reported a lignin model comprising phenolic β-o-4 type guaiacylglycerol-β-guaiacyl ether at 160–200 °C to investigate the pathways of lignin removal using high boiling solvent (HBS) [15]. The results showed that the phenolic β-o-4 linkage was broken down into two or more fragments and the scission of β-o-4 linkage revealed a pseudo-first-order reaction rate. Gasper et al. studied the aerobic lignin removal from Eucalyptus globulus kraft pulp [16]. The reaction was catalyzed by Mn-molybdovanado-phosphate polyoxoanion (HPA-5-Mn$^{II}$), and the measured rates of lignin removal was a pseudo-first-order model. Kim and Holtzapple, treated corn stover with an excess of $CaOH_2$ under oxidative and non-oxidative conditions and at various temperatures (25, 35, 45 and 55 °C) [17]. The results revealed first-order reactions for the delignification of corn stoves. Mbachu and Manley [18] studied the ozonolysis of spruce wood in aqueous acid media at room temperature. The degradation rate of lignin in spruce wood also followed a first-order kinetics. In this paper, the first-order model for delignification reactions, proposed by Mbachu and Manley, was followed for simplicity. However, this was an unsuccessful attempt and modifications were made to the proposed model [18]. The purpose of the present study is to determine the degradation rate of lignin on wheat straw after ozonation and to go into the mechanism of ozone transportation into the wheat straw matrix.

## 2. Materials and Methods

The following section describes the materials, analysis instruments, experimental setup, experimental preparation, and ozonation reaction for this study.

### 2.1. Materials

Wheat straw (*Tritium sativum*) from two bales was brought from a farmhouse in Toronto. Deionized water was used to wash the wheat straw. In addition, 1% NaOH solution (Sigma Aldrich, St. Louis, MO, USA) and 72% of Sulfuric acid ($H_2SO_4$) (Sigma Aldrich) solution were used for soaking and hydrolysis, respectively.

### 2.2. Instruments

Comfort machine, model SM100 (Retsch Inc., Haan, Germany), was used to mill the straws. Oven of type No. 3605 and 1200 watts was purchased from Labline Inc to dry the residues of the wheat straw. A generator type GL-1 (PC1-WEDCO) was used to produce ozone from pure oxygen. A spectrophotometer model 'S50', with an ultraviolet (UV) light detector was bought from Biochrom Libra to determine the lignin amount.

### 2.3. Experimental Setup

Figure 1a shows the experimental set up for ozonation reaction. The reactor is attached to ozone generator that can produce ozone at concentrations ranging from 6.5 to 65.5 mg from every liter of pure oxygen, which is supplied by oxygen cylinder. In fact, three different reactor designs were in contact with the incoming streamline. For instance, the simplest design is a column reactor of 3.5 × 25 cm size that uses fluidized-bed technique. The other two designs were the two-phase solid–gas fluidization reactor and the three-phase solid–water–gas fluidization reactor. Figure 1b shows the two-phase solid–gas fluidization reactor used in this investigation. It is a stainless-steel reaction chamber type that has an online spectrophotometer (257 nm) attached on the inlet line to the reactor. In addition, inlet and outlet control valves are fixed at the top of the reactor vessel. The lower section of the chamber holds a diffuser. Finally, the vent line is positioned at the top of the catalytic ozone destroyer vessel. In our previous study, we reported more details of this reactor [19].

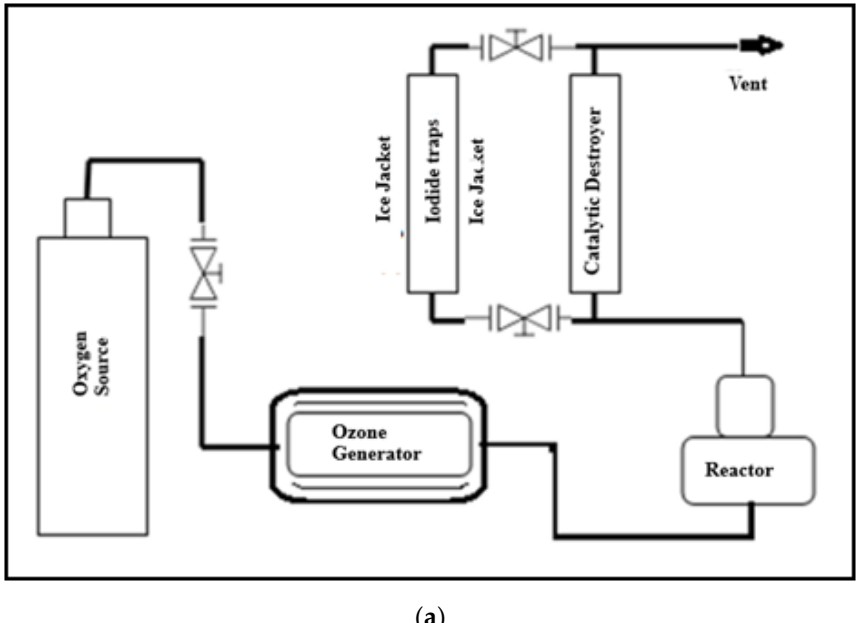

(**a**)

**Figure 1.** *Cont.*

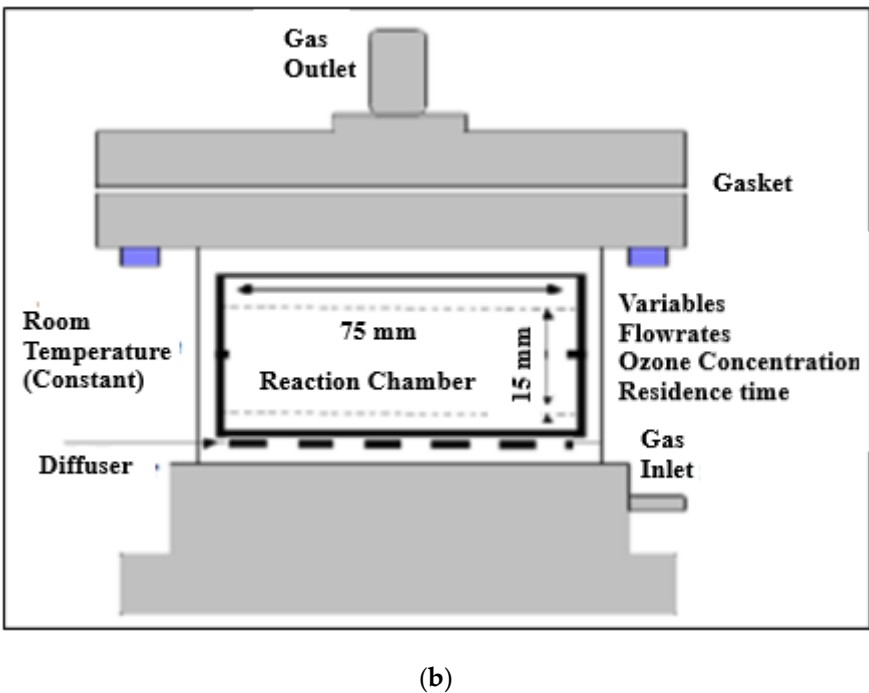

**(b)**

**Figure 1.** (**a**). Experimental setup for lignin removal, (**b**) ozonation reactor Reprinted with permission from Ref. [19]. Copyright 2015, copyright SAGE Publications.

### 2.4. Experimental Preparation

The preliminary treatment of samples was a three-step technique. First, dried wheat straw was soaked in 1% NaOH aqueous solution for 12 h. The resultant aqueous mixture was filtered using binder free-glass microfiber filter (grade 691) with a diameter of 9.0 cm and a pore size of 1.5 μm. The filtered residues were washed with deionized water until the pH value was around 7.0. Later, the washed residues were dried in an oven at 105 °C until a constant weight was obtained. Next, the dried sample was sprayed with water three times by a spray bottle. The sample was wrapped in aluminum foil to avoid water loss, then placed in covered Petri-dishes, and later wrapped in cellophane and stored at 4 °C for 24 h. A whole day was set out to allow the water to properly permeate into the wheat straw. The last step was to take out the moist sample from the wraps and ozonate it. The wheat straw used for ozonation had moisture contents of 1.6–1.8% wt. wt$^{-1}$.

### 2.5. Ozonation Reaction

After the sample preparation was done, the ozonation was performed after the wheat straw was charged into the reactor with a simple column design. The particles of the wheat straw began to stick on the reactor walls. The reaction was not satisfactory due to an insufficient reactant interaction, which meant that large amounts of the sample were wasted. Therefore, we decided to proceed with the two-phase solid—as fluidization reactor (Figure 1b) where maximum contact between the reactants was observed. The concentration of ozone coming in the reactor was spontaneously quantified by the online spectrophotometer at a wavelength of 257 nm. The diffuser below the reaction chamber disseminated the ozone before injection while the non-reacted ozone was allowed to go out from the reactor, and passed through a catalytic destroyer and out to the vent. The cylinder outlet pressure was 32 psig, and the reaction was carried out at room temperature (22 °C) during the entire ozonation reaction. After the desired period of ozonation, the treated samples were purged with oxygen for 5 min to eliminate any chance of continuous reaction for the ozone entrapped in cell walls. Later, the samples were degassed overnight before being tested for residual lignin using the Klason method.

## 2.6. Klason Lignin Constituents

The Klason method was used to determine the lignin contents. At the beginning, the ozone-treated samples were hydrolyzed using 15 mL of 72% $H_2SO_4$ at 4 °C for 20 min, and at 22 °C for 2 h, and finally boiled with 3% $H_2SO_4$ for 4 h. The samples were mixed every 15 min at room temperature perform hydrolysis. This is important to provide a homogenous contact of acid with particles (wheat straw). After cooling, the solution was filtered using the glass filter. The solids remaining at the end of hydrolysis were dehydrated overnight at 105 °C. The acid-insoluble lignin known as Klason lignin is matched by the difference in the number of dehydrated solids and ash. The experimental procedure reported by Bunzel et al., was followed to determine the weight of the ash by heating solids to 570 °C for 3 h [20]. Moreover, Ajao et al. reported the experimental procedure to determine the acid-soluble lignin content in the filtrates using UV/Visible spectrophotometer, in our case at a wavelength of 205 nm [21].

## 2.7. Measurement Reliability and Replication

It is important to make sure that the experimental data are reliable and each run was carried out with minimal errors. In this paper, the results were triplicates made for each ozone supply rate and the averages were presented on the plots. The experimental data were tested on different models for reproducibility purposes.

## 3. Results and Discussion

In general, the lignin constituents of wheat straw samples began to steadily decrease as the ozonation reaction started. Figure 2 depicts the amount of lignin that was eliminated at various ozone flow rates, ranging from 6.55 to 150 mg min$^{-1}$. The ozone in the inlet ozone–oxygen streamline was 2% wt. wt.$^{-1}$, and the reaction time (i.e., time of contact between ozone and wheat straw) ranged from 1 to 120 min. In the beginning, a very fast degradation rate was observed, as the ozone had easy access to the lignin. At 65 mg min$^{-1}$ ozone supply, the results showed that 65% and then over 90% of lignin was removed in the first 10 min and 110 min, respectively. In fact, the reaction rate varied with respect to the ease of access to lignin content in the pretreated samples. For instance, after a 90 min reaction, there was a clear reduction in the delignification rate because most of the lignin within accessible regions was consumed. Moreover, after a 60-min reaction, approximately 88% of the lignin was removed. Furthermore, the results also revealed that only a 3–5% change in lignin occurred over a long period of time (ranging between 60 to 120 min). The findings showed that 90% of the insoluble lignin was removed from the total lignin content of natural wheat straw after 90 min of contact time at the ozone flow rate of 150 mg min$^{-1}$. After a 120-min reaction, the amount of lignin removed was almost the same for ozone flow rates of 30 and 65 mg min$^{-1}$. Later, the reaction time was increased from 120 to 180 min and no significant difference in lignin removal was detected for all ozone flow rates. Interestingly, the remaining 8–9% lignin resisted removal under the current experimental conditions.

Whether it is possible to completely remove lignin from wheat straw is a subject of debate. Some of them claim that all the lignin constituents can react with ozone, while others say that a certain amount of lignin is unable to react [14,21–23]. Our experimental study supports the belief that a certain amount of lignin is left behind after ozonation. Kristensen et al. reported the use of scanning electron microscope (SEM) images to present the accessibility routes for ozone [24]. The images were taken after ozone pretreatment of wheat straw and showed lignin shells, which are hollow parts containing cellulose. Our findings stipulate that, under certain experimental conditions, ozone would not reach the deepest lignin sites. This is not to say that ozone cannot diffuse to the desired point; rather, ozone may damage the component of interest, in this case, cellulose.

In this study, a pseudo-first-order model and pseudo-second-order model were used for the first time to investigate the kinetics of ozonation on wheat straw that has been carried out in a solid–gas reactor. For each order of reaction, the coefficients were computed

as a number between 0 and 1 and represented as $r^2$ to evaluate the delignification process. In general, the higher the value of the determined coefficient, the better the fit is thought to be. Mbachu and Manley [18] presented the linear mode of a pseudo-first-order model, as shown in Equation (1):

$$\log_{10} \frac{L_0}{L_t} = \frac{1}{2.303} K_1 t \tag{1}$$

where $L_t$ and $L_0$ represent the amount of lignin removed (mg g$^{-1}$) from wheat straw at any time '$t$' (min), and the initial amount of lignin in the biomass, respectively. $K_1$ is the reaction rate constant for a pseudo-first-order model. Figure 3 shows the linear mode plot of the pseudo-first-order model for the delignification of wheat straw under different ozone flow rates of 30, 65 and 150 mg min$^{-1}$.

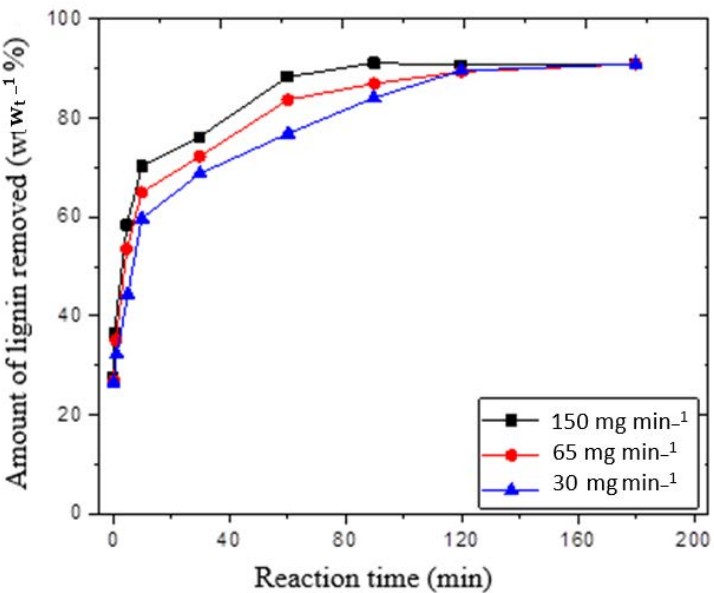

**Figure 2.** Lignin removal by reaction with ozone.

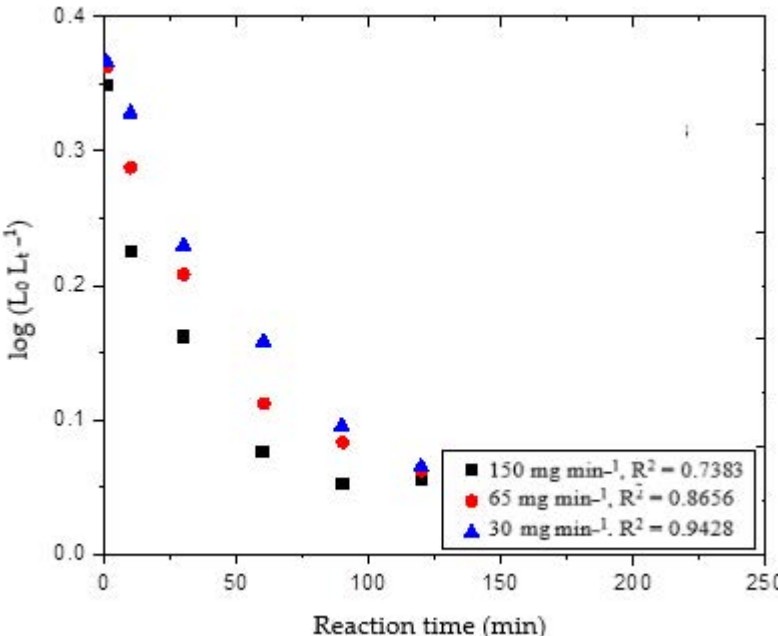

**Figure 3.** Delignification trends for the pseudo-first-order reaction model.

Theoretically, the plots should be straight lines while the number of experimental datapoints deviated from the ideal lines; therefore, the correlation coefficients were in the range of 0.74 to 0.94. The $r^2$ value indicates that the variation in reaction time can explain approximately from 74 to 94% of the variation in the concentration of lignin removal [log $(L_o L_t^{-1})$], and that the variables have a moderate degree of correlation [25]. The correlation factor elucidated that the reaction model was not suitable for the present study. The slope and intercept of these plots were used to determine the reaction constant $K_1$, and the amount of lignin removed at any given time $L_t$. Table 1 presents a comparison of the predicted and experimental values of $L_t$ for the corresponding flow rates. Additionally, we found that lignin degradation rate occurred approximately 1.25 times faster at an ozone supply of 30 mg min$^{-1}$ than at a flow rate of 150 mg min$^{-1}$. However, the pseudo-first-order model showed poor fitting, with correlation values of 0.74, 0.87 and 0.94 for 30, 65 and 150 mg min$^{-1}$, respectively. Based on the results, the pseudo second-order-model was next chosen for the ozone-lignin reaction.

**Table 1.** Parameters from pseudo-first-order reaction.

| O$_3$ Supply (mg min$^{-1}$) | L$_{max}$, Predicted | L$_t$ Actual | k |
|:---:|:---:|:---:|:---:|
| 30 | 137 | 118 | $5.76 \times 10^{-03}$ |
| 65 | 140 | 129 | $5.30 \times 10^{-03}$ |
| 150 | 145 | 139 | $4.61 \times 10^{-03}$ |

Iribarne and Schroeder suggested lignin removal model from kraft pulp using oxygen [26]. In Equation (2), a power law relationship is presented, which describes the kinetics govern lignin removal by oxidation, and also indicates the impact of the main reaction parameters such as caustic concentration, oxygen concentration (pressure) and process temperature.

$$\frac{dL}{dt} = k_1 e^{\frac{-E}{RT}} \left[OH^-\right]^\alpha \left[O_2\right]^\beta \left[L\right]^n \tag{2}$$

where $E$ is the activation energy; $R$ is the ideal gas constant. Moreover, the reaction constants are represented by $\alpha$, $\beta$ and $n$, while $k$ is the reaction rate coefficient, which depends on the reaction temperature in terms of the Arrhenius law, as shown in Equation (3):

$$k = k_1 \, e^{\frac{-E}{RT}} \tag{3}$$

When no acid or alkali were used in the system, the change in $[OH^-]$ was insignificant. The supply of ozone was deemed excessive; therefore, Equation (2) turns into Equation (4).

$$\frac{dL}{dt} = k \, [L]^n \tag{4}$$

Equation (5) is the pseudo-second-order model modified from Equation (4) and used to study the kinetics of lignin removal.

$$\frac{dL}{dt} = -kL^2 \tag{5}$$

where $k$ is the delignification reaction rate constant (mg$^{-1}$ min$^{-1}$).

The amount of lignin reacted and removed by ozone in wheat straw at any time $t$ is represented as $L_t$, while the maximum amount of lignin available for removal is $L_{max}$, which is equal to $(L_o - L_r)$. $L_r$ represents the amount of lignin that could be removed under experimental conditions. The driving force for the delignification of wheat straw is related to $(L_{max} - L_t)$. As a result, Equation (5) is rearranged by separating the variables, as given in Equation (6):

$$\frac{dL}{L^2} = -k \, dt \tag{6}$$

In addition, the integration of Equation (6) for the boundary conditions where t is 0 to *t* and L is $L_{max}$ to $(L_{max}-L_t)$ gives Equation (7):

$$\int_{L_{max}}^{L_{max}-L_t} \frac{dL}{L^2} = -k_2 \int_0^t dt \tag{7}$$

$$\left[-\frac{1}{L}\right]_{L_{max}}^{L_{max}-L_t} = -k_2\, t \tag{8}$$

Again, rearrange the variables in Equation (8). This results in the solution of delignification kinetics, where Equation (5) has the mode of pseudo-second-order model and is presented as follows in Equation (9):

$$\frac{t}{L_t} = \frac{1}{k_2 L^2{}_{max}} + \left(\frac{1}{L_{max}}\right) t \tag{9}$$

$k_2$ is the reaction rate constant for the pseudo-second-order.

Figure 4 shows the plot of $t/L_t$ versus t for wheat straw. The values of $L_t$ and $k_2$ are obtained by the slope and intercept.

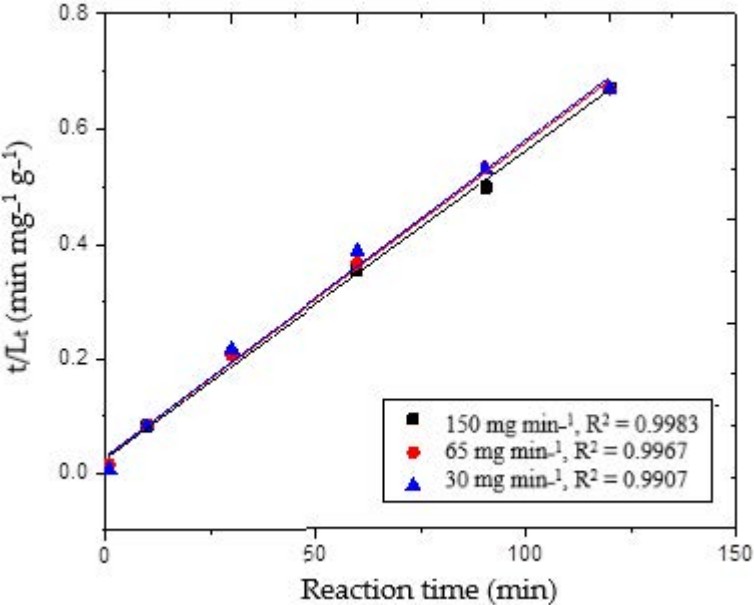

**Figure 4.** Delignification trends using the pseudo-second-order reaction expression.

The correlation coefficient ($r^2$) values obtained from the pseudo-second-order model were 0.9883, 0.9940 and 0.9977 for ozone supply of 30, 65 and 150 mg min$^{-1}$, respectively. We found that the $r^2$ value obtained here is greater than the one in the pseudo-first-order model. In other words, the lower the error in the prediction of $L_{max}$, the kinetics of ozone reaction with lignin in wheat straw likely follows a pseudo-second-order reaction model. Table 2 shows a further test that was conducted to test the model prediction, in which an ozone reaction with wheat straw was performed for 60 min. This indicated that the prediction error decreased from 4.5 to 0.5% when the ozone supply increased from 30 to 150 mg min$^{-1}$.

It is well-known that wheat straw is a biological material made up of various components. Therefore, it is necessary to understand thoroughly the transport mechanism of the delignification reaction in this complicated pattern. First, the kinetics of delignification are explained by the Elovich kinetic model, which assumes that the wheat straw–ozone reaction takes place on the surface. The model holds true when the reversible rate of reaction is negligible. Equation (10) expresses the linear mode of the Elovich kinetic model [27,28]

$$L_t = \frac{1}{\beta} \ln(\propto \beta\, t) = \frac{1}{\beta} \ln(\alpha\beta) + \frac{1}{\beta} \ln t \tag{10}$$

where the initial delignification rate is $\alpha$ (mg g$^{-1}$ min$^{-1}$) and the parameter $\beta$ represents the extent of surface coverage by ozone (g mg$^{-1}$). To simplify the Elovich equation, it is assumed that $\alpha\beta t \ll 1$. The model is useful in describing chemisorption on highly heterogeneous adsorbents such as agricultural waste materials (wheat straw). On the $L_t$ plot against $\ln (t)$, shown in Figure 5, the kinetic results were linear. The delignification curve showed poor suitability for the $r^2$ values, which varied from 0.84 to 0.94. From these results, it is clear that the delignification of wheat straw by ozone is affected by more than just a surface reaction; other reactions may also be involved.

**Table 2.** Amount of lignin removed in comparison with pseudo-second-order reaction.

| O$_3$ Supply (mg min$^{-1}$) | $L_t$, Predicted | $L_t$, Experimental | Error (%) |
|:---:|:---:|:---:|:---:|
| 30 | 126 | 118 | 4.35 |
| 65 | 132 | 129 | 2.27 |
| 150 | 138 | 139 | 0.70 |

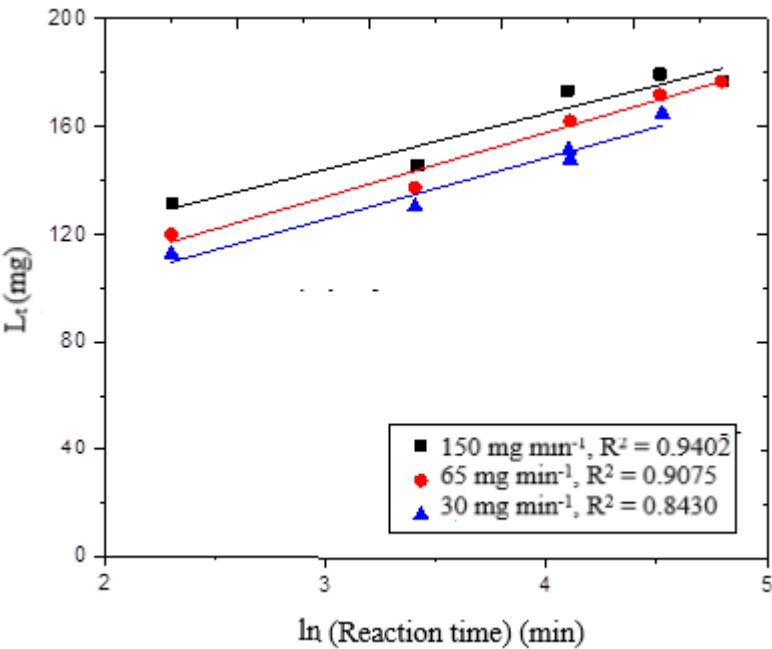

**Figure 5.** Elovich model for delignification of wheat straw using ozone.

The experimental data were further examined using the intra-particle diffusion model. Alkan et al. found that, in many chemical reactions, solute uptake varies proportionally with $t^{0.5}$ rather than the total reaction time [29], as shown in Equation (11):

$$L_t = k_3 t^{0.5} + C \tag{11}$$

where C is the intercept (mg g$^{-1}$) and $k_3$ is the rate of intra-particle diffusion (mg g$^{-1}$ min$^{-0.5}$). According to the model, if the ozone reaction with wheat straw was solely controlled by the intra-particle diffusion, the plot of $L_t$ against $t^{0.5}$ would be linear. This was not the case here, since the intra-particle diffusion was one of the factors contributing delignification reactions.

Figure 6 depicts a single-stage straight line. We think this may be attributed to the rapid covering of the accessible sites on the wheat straw surface. The correlation coefficient for intra-particle diffusion into wheat straw is a straight line with an $r^2$ greater than 0.92. This means that while the intra-particle diffusion was the primary reaction, it was not

the only one that contributed to wheat straw delignification. At 30, 65 and 150 mg min$^{-1}$ ozone supply, the predicted value of lignin removed from $L_t$ was evaluated when wheat straw was reacted with ozone for 60 min. The estimated error varied from 0.8 to 7%. The data were further examined in order to reveal more about pore diffusion during the delignification process.

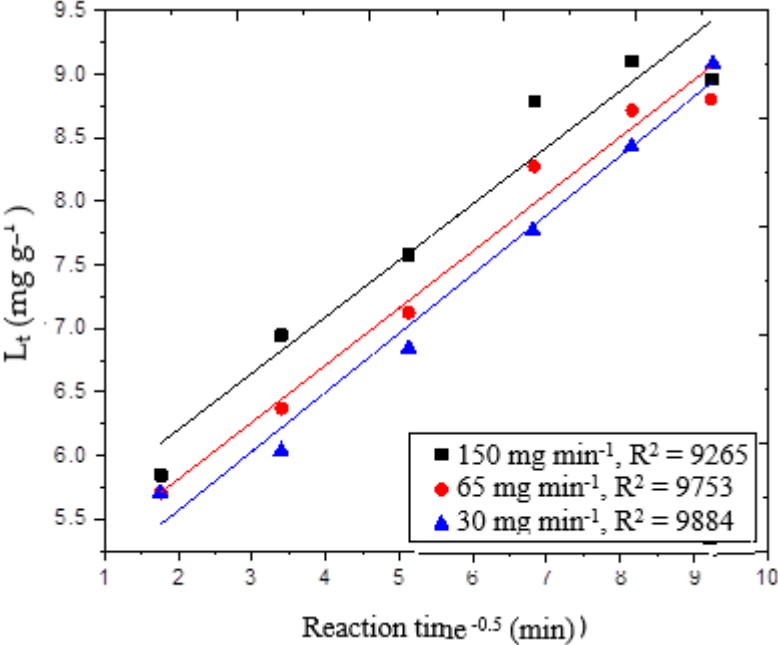

**Figure 6.** Delignification of wheat straw by diffusion of ozone.

An interesting finding is that at the high ozone supply of 150 mg min$^{-1}$, ozone may react with lignin on the surface and through intra-particle diffusion. Moreover, at a low ozone supply of 30 mg min$^{-1}$, ozone may react with the sites of contact, then move to the next site, and so on, until it loses the ability to react or accesses fewer sites.

Figure 7 shows an SEM image of pores on the surface of wheat straw. These pores help ozone to diffuse and react with available lignin sites within wheat straw.

Few researchers have studied the reactivity of lignin compounds with ozone and cover the kinetic models that describe various reaction mechanisms [30,31]. On the basis of the literature survey and the experimental results presented in various Tables and Figures in this article, it can be concluded that various reaction pathways may exist due to the existence of various morphological patterns of wheat straw components (lignin, cellulose and hemicellulose), inaccessible lignin sites hidden beneath the layers of other components, the cross-linking of reaction products, and reaction types (physical, chemical). Furthermore, we found that there are various lignin removal pathways involving ozonation, such as adsorption, surface reaction, surface diffusion, and/or intraarticular diffusion. On the basis of this study, the author proposes an ozonation mechanism of lignin removal: (i) ozone adsorbed onto the surface of wheat straw that chemically reacts with the available reaction sites at the surface, (ii) one molecule of ozone reacts with one site; another molecule reaches to the next site through intra-particle diffusion, and (iii) ozone moves through the pores (Figure 7) in the wheat straw surface (pore diffusion occur) and reacts with other available sites as the lignin is removed. Further studies are required to conduct a physical evaluation of the proposed mechanism.

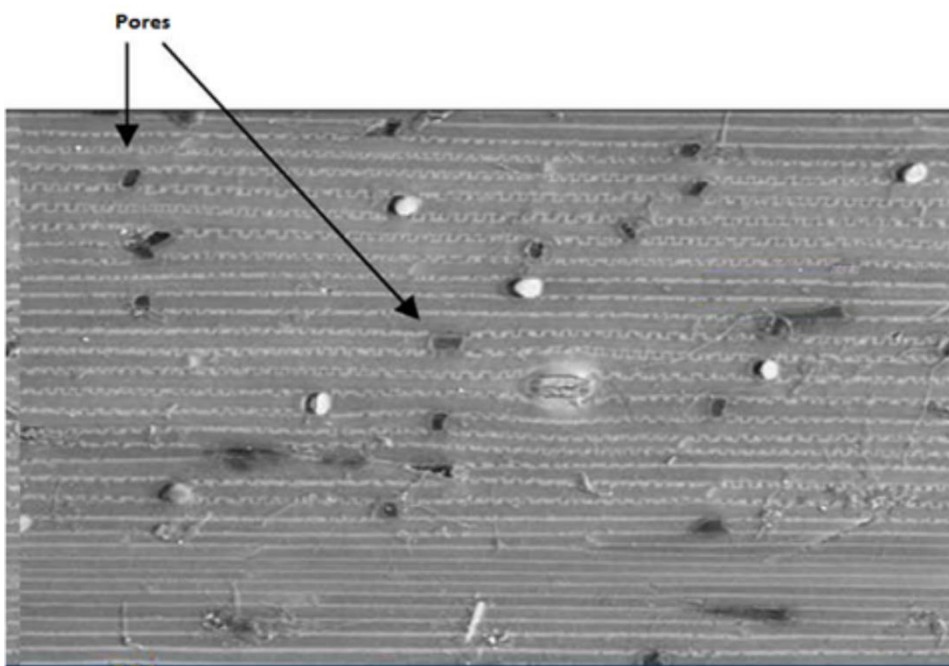

**Figure 7.** Pores in wheat straw surface.

## 4. Conclusions

This study successfully reported on the removal of lignin from wheat straw using ozone treatment in the two-phase solid–gas fluidization stainless steel reactor. A small amount of lignin is usually found deep within the biomass matrix, and it is difficult for ozone to reach that far end without damaging the cellulose. The key achievement of this study can be summarized with the kinetics of ozone delignification for wheat straw, which, in this case, follows a pseudo-second-order reaction. This agrees quite well with the promising results of the correlation coefficient $r^2$ that equals 0.9977, which is greater than the value reported in the pseudo-first-order model. We also recommend that further research is conducted to explore the kinetics of adsorption and chemical reactions in the delignification of wheat straw by ozone. As a final observation, it is important to mention that the adsorption took place at the surface of the wheat straw, and through intra-particle diffusion and pore diffusion, ozone reaches lignin for reaction.

**Funding:** This research received no external funding.

**Data Availability Statement:** The datasets generated during and/or analysed during the current study are available from the corresponding author on reasonable request.

**Acknowledgments:** The author wish to thank Ryerson University and University of Wah for moral support.

**Conflicts of Interest:** The author declares no conflict of interest.

**Sample Availability:** Samples of the compounds are available from the authors for exchange.

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
