# Peer review of "Kinetics of Lignin Removal from the Lignocellulosic Matrix after Ozone Transportation"

_methane, doi:10.3390/methane1030014_

Round 1
Reviewer 1 Report
This study presents a new method to remove lignin from wheat straw (lignocellulosic) by using the ozonation technique. In a solid-gas two-phase fluidized stainless steel reactor, the degradation rate of lignin on wheat straw after ozonation was measured, the kinetics of lignin removal and the mechanism of ozone entering wheat straw matrix were studied. In this paper, The author provide a novel technique that is simple and environmentally friendly, which is a step-by-step treatment procedure. However, the article also has some problems that need to be modified.
I am listing below my major comments.
1. The clarity of the figure needs to be changed. Figure 1 is not clear enough
2. Some minor formatting errors need to be corrected, such as units in figure 2 should be lowercase, formula labels should be aligned, etc. The author should check the whole paper.
3. The description of figure 2 should be more clear, and the change of lignin removal at the corresponding ozone flow rate should be clearly described. For example, in line 199, the author stated that 90% of the insoluble lignin in the total lignin content of natural wheat straw was removed after contact time of 90 min, At what ozone flow rate?
4. In line 219 the author finally mentions that ozone may damage the component of interest, in this case, cellulose. What does that mean?
5. In line 242-243, the author described that the degradation rate of lignin was about 1.25 times faster when ozone water supply was 30 mg min-1 than when ozone water supply was 150 mg min-1. This result was different from that in Figure 2.
6. The experimental data of the author are limited, whether there is a positive effect on lignin removal when ozone flow continues to increase, and whether temperature affects the degradation effect.
7. In addition to kinetic calculation, it is suggested that the author infer some mechanism analysis from literature and experiments, and it is suggested to cite and refer to some literature. For example: Adsorption modeling, thermodynamics, and DFT simulation of tetracycline onto mesoporous and high-surface-area NaOH-activated macroalgae carbon. Journal of Hazardous Materials, 2022, 425: 127887.
8. The error data presented in table 2 is inconsistent with the data described in line 293
Author Response
All the suggestions you made are incorporated in the manuscript

Reviewer 2 Report
This paper is interesting, it deals about the kinetics of lignin removal from the lignocellulosic using ozone. My first comment is that English is poor, I got tired trying to help in the correction of English and I cannot understand why the English is poor if two co-authors have affiliation in Toronto University.
The second comment is that the paper does not have nothing exciting, finally, the main achievement is to fit experimental data to a second order model.
The third comment is that there are statements without due justification, and in this way they cannot be accepted.
It seems the material presented in this paper is an undergraduate thesis, and the paper can be modified after major modifications.
As a guide, I provide the following observations.
Page 1, line 26.- “Liquid fuels are generally a lignocellulosic biomass wastes that 26 used to produce energy”….please correct…”Liquid fuels are generally a lignocellulosic biomass wastes that are used to produce energy”
Page 1, line 27.- “They originally found in the materials of agricultural or animal.”….please correct….”They originally were found in the agricultural or animal materials.”
Page 1, line 39. “to reduce the particle size to enhance the surface area of substrate…”….”to reduce the particle size to increase the surface area of substrate…”
Page 2 , line 55.- “ozonation reaction on wheat Straw”…Please correct...” ozonation reaction of wheat Straw”
Page 2, line 56.- “it is better to make the wheat straw wet before the start 56 of ozonation”…..please correct….”it is better to wet the wheat straw before the start 56 of ozonation”
Page 2, line 63.- “ Our preliminary work showed if the surface is not cleaned there will be no reaction or insignificant reaction of ozone.”….please correct…..”. Our preliminary work showed that if the surface is not cleaned, there will be no reaction”.
Page 2, line 85 “…reported lignin model consist of phenolic”…..please correct…..”reported lignin model consisting of phenolic”
Page 2, line “the first order model for delignification reactions, which proposed by Mbachu…”…..please correct…..”the first order model for delignification reactions, proposed by Mbachu...”
Page 3.- line 101.- “ozonation and investigates the mechanism of ozone transportation….”….please correct….”ozonation and to investigate the mechanism of ozone transportation”
Page 3.- line 113.- “was used for dedusted and milled the 113 straws….”….please correct….”was used to dedust and mill the straws.”
Page 3, line 116.- “A spectrophotometer of S50 model operated by ultraviolet (UV)…”….please correct….”A spectrophotometer model S50, with a detector ultraviolet (UV)”
Page 4, figure 1-a, “catalytic destrover”…..please correct….”catalytic destroyer”…….what does it means “catalytic destroyer”?
Page 3.- line 119.- “The reactor is at- 119 tached to ozone generator that can produce ozone…”….please correct….” The reactor is connected to ozone generator that can produce ozone…”
Page 3, line 125.- “Figure 1b present the two-phase…”…..please correct…..” Figure 1b presents the two-phase…”
Page 4, line 140.- “NaOH aqueous solution and for 12 h….”….please correct…..” NaOH aqueous solution for 12 h….”
Page 4.- very end, “The last step was taken out the mois- 148 tened sample from the wraps and promptly subjected to ozonation…”….please correct ….” The last step was to take out the wet sample from the wraps and ozonate it….”
Page 4, very end.- “The wheat straw used 149 for ozonation was having moisture contents of….”….please correct…..” The wheat straw used for ozonation had moisture contents of…..”
Page 5, line 159.- “The diffuser below the reaction chamber was disseminated the 159 ozone before injection while the non-reacted ozone was departed the reactor, then passed 160 through a catalytic destroyer and out to the vent…”….please correct…..” The diffuser below the reaction chamber disseminated the ozone before injection while the non-reacted ozone was removed from the reactor, then passed it through a catalytic destroyer and then out to the vent”
Page 5, line 170.- “The samples were mixed every 170 15 min at room temperature hydrolysis.”….something is missing….Probably….”The samples were mixed every 15 min at room temperature to carry out hydrolysis….”
Page 5.- line. 191- “At 65 mg min-1 191 ozone supply, the results showed that 65% and then plus 25% of lignin amount were re- 192 moved in the first 10 min and 110 min, respectively…”…..please correct…..”At 65 mg min-1 ozone supply, the results showed that 65% and then plus 90% of lignin amount were removed in the first 10 min and 110 min, respectively”
Page 5, line “Furthermore, the results also revealed that only a 3-5% change in lignin amount over a long period of time…”…..please correct…..”Furthermore, the results also revealed that only a 3-5% change in lignin, with respect to 90%, over a long period of time…”
Page 6, line 201 “and there was no significant difference in the removal of lignin between 90 to min….” this is repetition of the text in page 5, very end.
Page 6, line 212 “Our vigorous experimental study supports the belief of a 212 certain amount of lignin left behind during ozonation…”….please correct…..”Our experimental study supports the belief of a certain amount of lignin left after ozonation…”
Page 6, line “Our findings 216 cast a light on the fact that under certain experimental conditions,…”….please correct…” Our findings indicate that under certain experimental conditions,…”
Page 7, line 235.- “The plot was linear, and the correlation coefficient was in range of 0.74 to 0.94….”
This is not true, the plots are not linear for any of the ozone concentrations 30, 65 and 150 mg/min. All the profiles look like parabolas.
Page 7, line 238.- “The correlation factor elucidated that the reaction model was suitable for the present study”….This is not true, if the correlation coefficient is 0.74 then the results are not represented correctly for a first order model.
Page 7, line 244.- “However, the pseudo first order model showed poor fitting, with correlation values of 0.74, 0.87 and 0.94 for 30, 65 and 150 mg min-1 , respectively….” The authors have here a contradiction, due they said at the beginning of the paragraph that the reaction model was suitable.
Page 7, line 249.- “which describes the 249 kinetics govern lignin removal by oxidation,….”….please correct….”which describes the kinetics governing lignin removal by oxidation,…”
Page 8, line 265.- “where k is the delignification reaction rate constant [mg/(mg min)]. Please revise the units
The discussion about the superficial reaction and intra diffusion must be justified and extended, it is very concrete and it does not convince the reader.
Page 8.- I am not going to continue with the English, but still there are many many details along the rest of the paper.
Page 11, line “An interesting finding is that at high ozone supply of 150 mg min-1 , ozone may react with lignin on the surface and through intra-particle diffusion. Moreover, at low ozone supply of 30 mg min-1 , ozone may react with the site comes in contact, then moves to the next site, and so on until it loses the ability to react or access less sites….”….please explain the reasons for all of these statements. It is not clear why these affirmations.
Page 12.-line 343.- “In this study for example, it was found 343 that the reaction of ozone with wheat straw was a chemical reaction”….I consider this text is redundant, is it possible to have other type of reactions different to chemical reactions in this process?
Page 12, line 347, “Second, ozone reacted with one site and moves towards the next site through intra-particle difusión…”…..this is not intra particle diffusion, if the ozone reacts with one site, it is not ozone any more and it can not move to other site. Probably the authors mean superficial diffusion instead of intra – particle diffusion.
Page 12.- Line 362.- “and through intra-particle diffusion and pore diffusion ozone reaches to lignin for reaction….”…..What is the difference between intraparticle diffusion and pore diffusion?
Author Response
All the suggestions you have made has been added to the manuscript

Round 2
Reviewer 1 Report
Accept
Author Response
Thanks.
Reviewer 2 Report
The paper could be interesting and useful for the readers, but I see the paper contains only three experiments, at 30, 65 and 130 mg/min and based on them, the authors do statements not supported by the experimental data. It seems this material is an undergraduate thesis with few experiments and a very weak theoretical analysis. There are still issues with English. The paper could be published after major modifications suggested in the following list.
Page 1, line 14.- In abstract.- “The findings show variable levels of conformity against different selected models”….please remove this text, there is a repetition with the following text…”The kinetics of lignin removal did not agree with the experimental data until the suggested model is modified to a pseudo second order.”
Page 1, line “there has seen a huge growth for consuming liquid fuels”….please correct…”there has been a huge growth for consuming liquid fuels”
Page 2, line 56, “the wheat straw before the start of ozonation because”….please correct….”the wheat straw before the start of ozonation because…”
Page 2, line 69.- “Based on vigorous literature survey”….please correct….”Based on literature survey”
Page 7, line 240.- “Additionally, we have found that lignin degradation rate occurred approximately 1.25 times faster at ozone supply of 30 mg min-1 than at a flow rate of 150 mg min -1 .”…This is not true, the opposite is true, according to Table I.
It is not necessary to present the results for first order reaction, figure 3, due they are not good, just mention them.
There are still issues with English, I recommend the revision of English for a specialist. In this revision I am not indicating the details, it is very time consuming. I did this in the first revision.
The authors report results only for three experiments, and this is little experimental information to obtain conclusions.
Page 11,. Line 330.- From line 330 until the end of the paper, the paragraphs contain several statements not supported by the information provided in the paper.
Line 322.- “This means that while the intra-particle diffusion was the primary reaction, it was not the only one contributing to wheat straw delignification” Please explain why intra particle diffusion was the primary reaction?
Line 331.- “Moreover, at low ozone supply of 30 mg min-1 , ozone may react with the site comes in contact, then moves to the next site, and so on until it loses the ability to react or access less sites.” Please explain why
Line 339.- On this basis, we conclude that various reaction pathways may exist because of the existence of various patterns from different species, cross-linking of reaction products, and inaccessible sites of lignin hidden beneath the layers of cellulose. In this study for example, it was found that the reaction of ozone with wheat straw was a chemical reaction. This reaction occured through adsorption. Furthermore, our findings also showed three possible reaction pathways in this paper. First, ozone adsorbed onto the surface of wheat straw and chemically react with the available reaction sites at the surface. Second, ozone reacted with one site and moves towards the next site through intra-particle diffusion. And third, ozone moves through pores in the wheat straw surface (pore diffusion) and reacts with other available sites.”, there are a lot of statements without justification, and not supported by the experimental information. Please explain why these affirmations.
Author Response
Dear Reviewer,
Please find the corrections attached.
Thanks
